Citations and the h index of soil researchers and journals in the Web of Science, Scopus, and Google Scholar

Minasny Budiman 1 budiman.minasny@sydney.edu.au
Hartemink Alfred E. 2
McBratney Alex 1
Jang Ho-Jun 1
1 Department of Environmental Sciences, Faculty of Agriculture & Environment, The University of Sydney , Australia
2 Department of Soil Science, FD Hole Soils Lab, University of Wisconsin – Madison , Madison, WI , USA
Johnson Stephen
Electronic publication date: 2013 Oct 22
Publication date: 2013
Volume: 1
Electronic Location ID: e183
Received 2013 Sep 9; Accepted 2013 Oct 1
Copyright: © 2013 Minasny et al.
Copyright year: 2013
Copyright holder: Minasny et al.
License: This is an open access article distributed under the terms of the Creative Commons Attribution License, which permits unrestricted use, distribution, and reproduction in any medium, provided the original author and source are credited.
License URL: https://creativecommons.org/licenses/by/3.0/

Keywords: Soil science, Bibliometrics, h index, Impact factor, Citations, Transfer functions

Funding: There was no funding for this work.

==============================
Citation metrics and h indices differ using different bibliometric databases. We compiled the number of publications, number of citations, h index and year since the first publication from 340 soil researchers from all over the world. On average, Google Scholar has the highest h index, number of publications and citations per researcher, and the Web of Science the lowest. The number of papers in Google Scholar is on average 2.3 times higher and the number of citations is 1.9 times higher compared to the data in the Web of Science. Scopus metrics are slightly higher than that of the Web of Science. The h index in Google Scholar is on average 1.4 times larger than Web of Science, and the h index in Scopus is on average 1.1 times larger than Web of Science. Over time, the metrics increase in all three databases but fastest in Google Scholar. The h index of an individual soil scientist is about 0.7 times the number of years since his/her first publication. There is a large difference between the number of citations, number of publications and the h index using the three databases. From this analysis it can be concluded that the choice of the database affects widely-used citation and evaluation metrics but that bibliometric transfer functions exist to relate the metrics from these three databases. We also investigated the relationship between journal’s impact factor and Google Scholar’s h5-index. The h5-index is a better measure of a journal’s citation than the 2 or 5 year window impact factor.

Introduction

Scientific impact measures are increasingly being used for academic promotions, grant evaluations and evaluation of job vacancy candidates. They are also being used for the evaluations of university departments and research centres. Traditionally, the impact factor of a journal has been used – a metric developed by Garfield (1955) whereby the citations and number of papers published over a given period are divided. For most journals it shows considerable inter-annual fluctuation and it provides no information on individual papers nor individual authors. Since 2005, the h index has been used as an index for quantifying the scientific productivity of scientists based on their publication record (Hirsch, 2005). It is a personal index and provides information on the number of publications of an author and the number of citations: A scholar with an index of h has published h papers with at least h citations each. The h index can also be calculated for journals, departments, universities or countries.

The three widely used bibliometric databases for analysis and evaluations of citations and the h index are Web of Science (Thomson Reuters), Scopus (Elsevier), and Google Scholar. Some papers have compared citations between these three databases. Although Google Scholar and Scopus seem to provide higher numbers of citations (Falagas et al., 2008), there is mixed information on the h index. For example, Bar-Ilan (2008) compared the h index for 47 highly-cited Israeli researchers across the three databases and concluded that the results from Google Scholar are considerably different from Web of Science and Scopus. Mingers & Lipitakis (2010) looked at 4,600 publications from three UK business schools, and found that Web of Science poorly covers the management discipline compared to Google Scholar. De Groote & Raszewski (2012) examined 31 faculty members from nursing faculty in the Midwestern USA, and concluded that more than one database should be used to calculate the h index. They further recommended that since the h index rankings differ among databases, comparisons between researchers should be done only within a specified database.

The difference between the three databases has been fairly well established and the three databases will calculate different citations and h indices. As far as we know, the relationships between the three databases have not been investigated and derived. The aims of this paper are therefore: (i) to compare citations and h index across the three databases, (ii) to derive transfer functions to convert metrics from one database to the others, and (iii) to compare impact factors for journals and the h index. Hereto we have compared the data from 340 researchers and 31 journals. Since we are soil scientists, we have used only soil researchers and journals in this study.

Soil science is a study of soil as a natural phenomenon and resource (Brevik & Hartemink, 2010). It is a relatively small discipline in terms of number of researchers, number of papers per annum, and citations. The IUSS (International Union of Soil Sciences) database lists about 50,000 soil scientists worldwide, but only a fraction of these are in research and actively publish, with a guesstimate of 5,000 to 10,000 publishing researchers. The “soil” topic has lower number of papers and citations when compared to other subjects of natural resources such as “air” and “water” (Minasny, Hartemink & McBratney, 2007). The number of published papers in 2011 according to Scopus with “soil” in the abstract and keywords is 39,504, with a rate of increase of about 2,000 papers per year. In comparison the number of papers in 2011 on “air” is 1.4 times larger and the number of papers on “water” is 3.5 times larger. The h index ratios for water, air, and soil (for the papers published in 2011) are 1.7, 1.3, and 1.0. Nevertheless soil is becoming more important with strong links to global issues of food security, biodiversity, land use change, and climate change (McBratney, Field & Koch, 2014). While this study only used soil researchers, the bibliometric results are illustrative to other agricultural, environmental, earth science and biology disciplines, and to small scientific disciplines in general.

Data and Methods

Google Scholar (GS) is a bibliographic database freely available from Google. It was introduced in 2004 and contains scholarly works across many disciplines and sources, including theses, books, reports, abstracts, peer-reviewed and non-reviewed articles, and web pages that are deemed scholarly. Google Scholar lists these automatically from its search engine activities (Harzing & van der Wal, 2009; Vine, 2006). An individual Google Scholar page was featured in 2012, where a researcher can create a webpage, with fields of interest. Google Scholar automatically searches and populates the individual’s publications, calculates and displays the individual’s total number of citations, h index, and i10 index. Scopus, or SciVerse Scopus, is a bibliographic database from Elsevier which contains abstracts and citations for academic journal articles, conference papers, and book chapters. Inclusion in the database is through the Scopus Content Selection and Advisory Board. Although its record goes back as early as 1823, its citations are reliable after 1995. The Web of Science is a bibliographic database from Thompson Reuters which only contained abstracts and citations for articles listed in the Web of Science indexed journals since 1900 (Harzing & van der Wal, 2009).

Data from researchers who listed their areas of interest as: “soil science”, “soil”, “pedology”, “soil physics”, “soil biology”, “soil chemistry”, “soil fertility”, “soil erosion”, “soil ecology”, and “soil carbon” were retrieved from the Google Scholar author pages. The same researchers were located in Scopus and the Web of Science. In Scopus, the ‘Author Identifier’ tool was used to locate the researcher. In the Web of Science, the author’s surname and first name’s initial was used, together with “soil” in the search subject. When the name and publication record were inconsistent across all three databases, the researcher was not included in our analysis. At the end, we collected data from 340 researchers and this included: number of total citations, h index, number of papers, and year of the first publication. These data were obtained for each researcher and from each of the three databases. The publications and citations are until June 2013.

Results and Discussion

Number of papers, citations and h index

Table 1 shows the statistics of h index, number of publications, number of citations, and year of the first paper for 340 soil researchers in the three databases. Our data encompass a wide range of researchers from early-career to well-established and highly-cited researchers. The database is much larger and more diverse than previous studies where a small and focussed group of researchers was used to compare citation metrics between the databases (e.g., Franceschet, 2010; Meho & Rogers, 2008; Patel et al., 2013).

Table 1 Descriptive statistics of publication indices from Google Scholar, Scopus and Web of Science database for 340 soil researchers.

	Number of
citations	Number of
papers	First year of
publication	h index	m (rate of h index
increase per year)	
Google Scholar						
Minimum	16	3	1953	1	0.09	
25th Quantile	266	32	1985	26	0.56	
Median	866	79	1993	15	0.85	
75th Quantile	2596	146	2001	26	1.18	
Maximum	49447	1159	2011	115	3.67	
Scopus						
Minimum	1	1	1955	1	0.03	
25th Quantile	116	14	1989	5	0.45	
Median	469	34	1996	11	0.71	
75th Quantile	1361	65	2004	19	1.00	
Maximum	28693	423	2011	70	2.87	
Web of Science						
Minimum	1	1	1957	1	0.06	
25th Quantile	76	10	1991	5	0.41	
Median	291	23	1998	10	0.67	
75th Quantile	945	48	2004	17	1.00	
Maximum	32837	424	2011	96	2.87	

The median number of papers for the 340 soil researchers ranged from 23 (Web of Science) to 79 (Google Scholar) with Scopus having intermediate values. The number of citations is also highest in Google scholar, with a median of 866 citations per author whereas it is 291 in the Web of Science. The h index and its annual increase are lowest in the Web of Science. This pattern holds for all of the metrics presented here: Google Scholar has the highest numbers and the Web of Science the lowest whereas the Scopus numbers are in between. Part of this may be the different types of publications included and also the periods of time covered by the 3 databases are slightly different.

A simple linear regression without intercept was performed between the citation indices of the three databases (Table 2). Google Scholar has on average 2.3 times more articles and 1.9 times more citations than the Web of Science. The Scopus database (all years) has 1.1 times more papers than the Web of Science but a similar number of citations compared to the Web of Science. Since the citations are more correct and complete after 1995, a revision was made to the relationship for post 1995 authors; it shows that Scopus has about 1.2 times more citations than the Web of Science. The 20% higher citations are consistent with the findings by Falagas et al. (2008) in the field of medicine. Similarly, for articles in medical journals, Kulkarni et al. (2009) found that Google Scholar and Scopus retrieved more citations compared to Web of Science (1.22 and 1.20 times respectively).

Table 2 Comparison of publication indices from Google Scholar (GS), Scopus and Web of Science (WoS).

	Standard error
of estimates	R 2	
GS no. papers = 2.33 WoS no. papers	0.06	0.797	
GS no. citations = 1.87 WoS no. citations	0.05	0.809	
GS h index = 1.44 WoS h index	0.02	0.956	
Scopus no. papers = 1.09 WoS no. papers	0.02	0.902	
Scopus no. citations = 1.03 WoS no. citations	0.02	0.867	
Scopus h index = 0.99 WoS h index	0.01	0.936	
Authors who started to publish after 1995			
Scopus no. papers = 1.11 WoS no. papers	0.03	0.900	
Scopus no. citations = 1.17 WoS no. citations	0.02	0.949	
Scopus h index = 1.11 WoS h index	0.02	0.954	

Figure 1 Relationship between the number of papers, the number of citations, and the h index of 340 soil researchers in the Web of Science (WoS), Scopus and Google Scholar (GS).

The relationship between number of papers and citations is scattered, especially for the number of papers (Fig. 1), but the relationships between h index values across the 3 databases appear to be quite linear. The h index in Google Scholar is on average 1.4 times larger than Web of Science, and the h index in Scopus (post 1995 authors) is on average 1.1 times larger than Web of Science. However, for pre-1995 authors, their Scopus h index is similar and sometimes can be smaller when compared to Web of Science. While Google Scholar contains more grey literature (informally published written material) and its citations may contain errors (Harzing & van der Wal, 2009), the h index appears to be quite robust and comparable with Web of Science and Scopus. This is due to the fact that h index does not vary greatly if the number of articles increases (e.g., book chapters or unrefereed articles). In addition, extra citations do not have a large effect on the h index, as once a paper has reached h citations additional citations to that paper do not affect its value (Franceschini, Maisano & Mastrogiacomo, 2013; Courtault & Hayek, 2008).

Our results are different from the study by Franceschet (2010) who evaluated 13 computer scientists from his university’s department and found that on average Google Scholar had five times more papers, eight times more citations and a three-fold larger h index. Our results from 340 soil researchers are more in line with De Groote & Raszewski (2012) who looked at 30 researchers from nursing and found that the h index from Google Scholar is 1.3 times larger than the Web of Science, and Scopus is 1.1 times larger than the h index in the Web of Science. Similar results were obtained by Meho & Rogers (2008) who evaluated 22 human–computer interaction researchers from the UK and found that the h index in Google Scholar is on average 1.6 times higher than Web of Science. Patel et al. (2013) compared publications and citations for 195 Nobel Laureates in Physiology and Medicine using the three databases. They found no concordance between the three databases when considering the number of publications and citations count per Laureate. However, the h index was the most reliably calculated bibliometric index across the three databases.

We calculated the Spearman’s rank correlation (ρ) of the h index of the 340 researchers from the three databases. The three databases show excellent correlation for the h index, with WoS and GS as having the largest concordance. Unexpectedly, the rank correlation in terms of no. citations and no. papers (Table 3) also indicates that the three databases are comparable. This implies that the ranking of individuals within a database is comparable with the other databases. Our correlation is also much higher compared to the 13 computer scientists studied by Franceschet (2010) who only obtained ρ = 0.65. We used a much larger dataset, and the GS data came from the page that was created by the researcher, thus the listed papers and citations are assumed to be more complete.

Table 3 Spearman’s rank correlation coefficient (ρ) of the h index of the 340 researchers using Google Scholar (GS), Scopus and Web of Science (WoS).

Variable	By variable	Spearman ρ	Prob > |ρ|	
WoS h index	GS h index	0.939	<.0001	
Scopus h index	GS h index	0.931	<.0001	
WoS h index	Scopus h index	0.922	<.0001	
WoS no. citations	GS no. citations	0.939	<.0001	
Scopus no. citations	GS no. citations	0.955	<.0001	
WoS no. citations	Scopus no. citations	0.945	<.0001	
WoS no. papers	GS no. papers	0.840	<.0001	
Scopus no. papers	GS no. papers	0.896	<.0001	
WoS no. papers	Scopus no. papers	0.905	<.0001	

The h index of soil researchers

In an earlier paper (Minasny, Hartemink & McBratney, 2007) we investigated the relationship between the h index of 228 soil researchers and found that the index was 0.7 times the number of years since the first publication (which we called scientific age, or t). That means if a researcher has been publishing for 10 years his/her h index should be about 7. We calculated this index using the Web of Science database, and now we repeated this using analysis to the 340 soil researchers in this study (which are different from the previous list) (Fig. 2). Although the data are scattered the relationship holds: h index=0.73× scientific age(R2=0.72).

The Web of Science database shows that the average rate of h index increase over time (m) is 0.7, with the lowest value of 0.06 and highest value of 2.9 (Table 1). The average m value for Scopus is 0.7 and for Google Scholar it is 0.8 (Table 1).

Figure 2 Relationship between the scientific age (t) of 340 soil researchers and the h index (Web of Science data).

McCarty & Jawitz (2013) evaluated the linear relationship between scientific age and h index for 4 disciplines and found the following mean m values of 0.83, 0.47, 0.43, and 0.36 for biochemistry, water, economics, and anthropology, respectively. Thus the trend of soil science is in between water and biochemistry.

For selected researchers, we tried to calculate the distribution of m (h index divided by the number of years since first publication) as a function of sub-disciplines in soil science. Table 5 shows the distribution of m for WoS and GS according to 6 sub-disciplines. It shows that h index varies between sub-diciplines, for WoS, soil biology, biogeochemistry and ecology have the highest m values (median of 0.8). This is followed by soil physics, soil fertility and management, soil geography and pedometrics, chemistry and lastly pedology (average m = 0.5). The order in Google Scholar is slightly different, but it is consistent in that soil biology has the highest m value and pedology is the lowest. Therefore within soil science, the sub-disciplines also vary in terms of h index. The citation ratios are: Soil biology, ecology and biogeochemistry, Soil management and fertility, Soil geography & pedometrics, Soil physics, Soil chemistry and mineralogy, Pedology 1: 0.9: 0.8: 0.8: 0.8: 0.6; respectively.

Although the number of citations for researchers across the three databases can be quite different, the relationship between the number of citations and the h index is quite consistent across the three databases (Fig. 3): h index=½n½(R2=0.95).

This relationship follows the function postulated by Hirsch (2005) where the number of citations is about 3 to 5 times h2, and it appears the h index follows an absorption-type relationship (Warrick, 2003), increasing rapidly at low numbers of citations with the rate decreasing with increasing number of citations.

Figure 3 Relationship between the number of citations and the h index of 340 soil researchers from 3 databases.

Black dots are data from Web of Science, green squares are from Scopus, and blue triangles are from Google Scholar.

Table 4 shows the relationship of the average number of papers and citations per year for the 340 soil researchers. This can be interpreted as: on average, a soil researcher produces 5 articles per year, 2 articles in international refereed journals, 1 in a conference proceedings, and 2 other unrefereed publications. The researcher receives 65 citations per year from journal articles, an additional 13 citations from conference proceedings and another 44 citations from other publications.

Table 4 Comparison of the h index over time using the data from 340 soil researchers in Google Scholar (GS), Scopus and Web of Science (WoS).

	Standard error of
estimates	R 2	
GS h index = 0.84 × year	0.02	0.745	
WoS h index = 0.73 × year	0.02	0.717	
Scopus h index = 0.73 × year	0.02	0.759	
GS no. papers = 5.5 × year	0.23	0.620	
WoS no. papers = 2.5 × year	0.12	0.567	
Scopus no. papers = 3.0 × year	0.12	0.656	
GS no. citations = 122 × year	9.1	0.344	
WoS no. citations = 65 × year	5.8	0.269	
Scopus no. citations = 78 × year	5.8	0.346	

Self-citations

A way to boost the h index is by self-citation. Hyland (2003) found that self-citation is 12% of all references in biology, engineering and physics, compared to 4% in sociology, philosophy, linguistics, or marketing. For soil science journals, we found a mean of 12% self-citations but it differs between the sub-disciplines (Minasny, Hartemink & McBratney, 2010). High rates of self-citation were accompanied by high journal impact factor ranking; China and the USA had the highest rates of self-citation whereas Egypt, Algeria, Ukraine, and Indonesia have low levels of self-citations in soil science (Minasny, Hartemink & McBratney, 2010).

So high rates of self-citation may influence the h index and the Scopus database allows calculation of the h index with and without self-citation. Self-citation here is the so-called diachronous kind (Lawani, 1982), which is self-citation from the citations received by the author. The other type is called synchronous which is more difficult to calculate, i.e., author’s self-referencing relative to the total number of references cited in a paper.

The relationship for the 340 soil researchers is consistent and the average h index without self-citation is about 12% lower (Fig. 4): h index without self-citation=0.88×h index (R2=0.97).

Figure 4 (A) Relationship between the h index with and without self-citation, (B) relationship between the scientific age of 340 soil researchers and percentage self-citation based on Scopus data.

We found a weak relationship between percentage of self-citation and scientific age (t) (Fig. 4). It suggests that some younger authors appear to have high rates of self citation as their works were not known widely and their citations mainly come from themselves, as the researchers mature their papers are more widely known and more external citations were gained thus a lower percentage of self-citations: Percent self-citation=42−5t0.5(R2=0.18).

Journal citations

We retrieved 31 Soil Science journal impact factors (IF) and other metrics from the 2012 Thompson Reuters Journal Citation Reports (JCR, released in June 2013). Google Scholar also has measures of the journal’s metric, the h5-index, which is the h index for articles published in that journal for the last five years. The list of journals for the soil science discipline in Google Scholar is slightly different from the Thompson Reuters Journal Citation Reports (JCR), and therefore we used the journals listed in JCR as the basis for comparison. We searched for the h5-index for the journals in Google Scholar metrics for 2012 (released July 2013).

Table 5 The distribution of m (h index divided by the number of years since first publication) according to sub-disciplines in soil science using the data from Google Scholar, Scopus and Web of Science, n is the number of samples, Q25 and Q75 refers to the lower and upper quartile.

	n	Min	Q25	Median	Q75	Max	
Google Scholar							
Pedology	25	0.09	0.37	0.56	0.83	1.67	
Soil chemistry	42	0.09	0.45	0.78	1.16	2.00	
Soil physics	67	0.21	0.56	0.79	1.00	2.55	
Soil geography & pedometrics	28	0.16	0.58	0.87	1.02	1.93	
Soil management & fertility	42	0.28	0.69	0.98	1.30	2.12	
Soil biology, ecology, biogeochemistry	88	0.23	0.78	1.01	1.65	3.67	
Web of Science							
Pedology	25	0.08	0.32	0.46	0.78	1.67	
Soil chemistry	42	0.06	0.32	0.63	1.00	1.67	
Soil geography & pedometrics	67	0.11	0.50	0.64	0.86	1.50	
Soil management & fertility	28	0.15	0.47	0.67	1.00	1.63	
Soil physics	42	0.17	0.46	0.72	1.00	2.14	
Soil biology, ecology, biogeochemistry	88	0.14	0.63	0.83	1.33	2.87	

Table 6 Spearman’s rank correlation coefficient (ρ) of the Google Scholar h5-index and impact factor (IF), no. papers, citations, and Eigenfactor metrics from Journal Citation Reports for 31 soil science journals. Cites is the number of citations in 2012 for papers that were published in 2007–2011.

Variable	By variable	Spearman ρ	Prob > |ρ|	
h5-index	Cites (5 years)	0.972	<.0001	
h5-index	Eigenfactor	0.970	<.0001	
h5-index	5 year IF	0.903	<.0001	
h5-index	2 year IF	0.870	<.0001	
h5-index	No. papers (5 years)	0.721	<.0001	

Table 6 shows that Google Scholar h5-index has a better correlation with the five year IF (impact factor) than the two year IF, and Fig. 5 shows the comparison between GS h5-index and the five year IF. While the h5-index and five year IF have a high rank correlation (ρ = 0.90), the ranking is different for different journals. The journals ‘Soil Biology and Biochemistry’ and ‘Plant and Soil’ both consistently ranked no. 1 and 2 in JCR and GS while other journals appear to be slightly different in their ranking (1 to 3 places difference). The top two journals are able to maintain a large number of citations relative to the number of papers they publish.

Figure 5 A comparison between 5 year Impact factor (IF) and Google Scholar h5-index for 31 soil science journals in 2012.

There are 4 journals that are ranked much higher ( > = 4 difference in rank) in Google Scholar compared to the IF: ‘Soil Science Society of America Journal’, ‘Journal of Plant Nutrition and Soil Science’, ‘Pedosphere’, and ‘Revista Brasileira de Ciência do Solo’. All these journals are published by national soil science societies (USA, Germany, China and Brazil). In the case of ‘Revista Brasileira de Ciência do Solo’ which ranked 12 in GS and 25 in JCR, Google Scholar includes more citations from non-English articles. Contrarily, there are four journals that are ranked much lower ( < = 4 difference in rank) in Google Scholar: ‘European Journal of Soil Science’, ‘Soil Use and Management’, ‘Journal of Soil and Water Conservation’, and ‘Soil Science’.

The Thompson Reuters Journal Citation Reports suffers from a miscalculation, for example, ‘Australian Journal of Soil Research’ was reported to have a 2 year IF of 3.443. This is a miscalculation, as the journal changed its name to ‘Soil Research’ in 2011, and the IF calculation for Australian Journal of Soil Research only accounts for papers published until 2010. ‘Soil Research’ was again listed as a separate journal in JCR. We have recalculated the actual impact factor for this journal in our analysis.

While there is a positive correlation between cites (citations in 2012 to papers published from the previous 5 years) and IF, we can see that there are 2 trends (Fig. 6A). For journals that published <700 papers between 2007 and 2011 (or on average less than 140 papers per year) IF tends to increase rapidly with increasing citations (1.2 increase in IF per 1000 citations). For the other 7 journals that published more than 700 papers, the slope is half as much (0.6 IF increase per 1000 citations). So there is a drawback for journals that publish more papers. Meanwhile the h5-index is mostly controlled by number of citations following an absorption relationship (Fig. 6B). Although the citations come from WoS, the h5-index still holds the square-root relationship supporting its robustness.

Figure 6 (A) Relationship between cites and 5 year Impact factor (IF), and (B) relationship between cites and Google Scholar h5-index for soil science journals in 2012. Cites is the number of citations in 2012 for papers that were published in 2007–2011.

Table 6 also shows that the GS h5-index is more correlated to the Eigenfactor metric compared to IF. The Eigenfactor metric is based on the Google PageRank algorithm which calculated the “influence” of the journal based on the citations weighted by the “quality” of the journal (Bergstrom, 2007). A citation from a highly cited journal will have a higher weighting than a lower cited journal. We showed that this metric is less vulnerable to self-citation than the impact factor (Minasny, Hartemink & McBratney, 2010). Interestingly, Google Scholar does not apply its PageRank for citations.

Vanclay (2007) and Courtault & Hayek (2008) established that the h index is robust and is relatively unaffected by grey literature and errors in citations such as in Google Scholar. Most of the errors (and distortions) in citation databases are found in the ‘long tails’ of the citation distribution and they tend not to affect the h index much. For the journals considered here, we calculated the ratio between h5-index and number of papers, and it shows that only 1–9% (median 5%) of the total papers that contributed to h5-index. In other words, less than 10% of the cited papers are influencing the h5-index. We also demonstrated that the h5-index keeps its relationship even when using WoS citations.

Harzing & van der Wal (2009) recommended the use of the GS h index for Management and International Business journals. We also concur that the h5-index is a better measure of a journal’s citation performance than the impact factor as it is more robust and less affected by citation manipulation. It is now acknowledged that there are ways of manipulating impact factor, which include self-citation, and editorials that listed references to previously published articles (Falagas et al., 2008). The h index is less sensitive to the increase in number of citations, while individual highly cited papers can artificially increase the impact factor. In addition, it only considers the top influential h papers in the journal, thus it does not penalise a journal for publishing a larger number of papers. Although h index can also be manipulated by self-citation, in order to increase the h index considerably, a journal has to be more tactical by increasing a significant number of citations to certain papers.

Conclusions

From this analysis the following can be concluded:

– There is a large difference between the number of citations, number of publications and the h index using the three different databases.

– On average, Google Scholar gives the largest number of publications, largest number of citations and the highest h index. The Web of Science gives the lowest averages.

– There are solid relationships between the h indices in these three databases.

– The h5-index has a correlation with the five year impact factor, but it is more robust and less affected by citation manipulation. It should be considered as an alternative to the journal’s impact factor.

This analysis has shown that the choice of the database affects the assessment of scientific impact for academic promotions, grant evaluations, job vacancy candidates or the evaluations of university departments and research centres. It is recommended that we should quote these bibliometric indices for all three databases as they reflect different types of publications. Web of Science uses mostly refereed journal articles, Scopus includes conference proceedings and book chapters, whereas Google Scholar includes other publications (including software). The established relationships between the databases (Table 2) can be used as bibliometric transfer functions by anyone interested in relating databases. We are not aware of whether these functions have been established for other scientific disciplines but assume they will be similar. As a test, we applied our function relating the h index of WoS and GS to the 30 nursing faculty data of De Groote & Raszewski (2012) and the function gives a good prediction with a Spearman rank correlation of 0.852. We envisage that these functions would work better for science than socio-economical disciplines. However, this needs to be investigated.

Although we focussed on the relatively small discipline of soil science, the reported researchers has a lot of cross-over with other disciplines, in particular earth science, agricultural science, biogeochemistry and ecology. Many researchers in ecology and microbiology work with soil as a medium, while they do not necessarily study soil as a natural body. Their contributions elevated the citations as compared to pure soil research. The trend of h index for soil researchers appears to be in between the water and biochemistry disciplines (McCarty & Jawitz, 2013). However, soil science publication rate (on average 2.5 papers per year per researcher) is lower compared to water and biochemistry (on average of 3.1 and 3.8 papers per year, respectively).

Supplemental Information

Supplemental Information 1 Bibliometric datasets of researchers and journals used in this study

Click here for additional data file.

Additional Information and Declarations

Competing Interests

Author Contributions

Budiman Minasny is an Academic Editor for PeerJ.

Budiman Minasny conceived and designed the experiments, performed the experiments, analyzed the data, contributed reagents/materials/analysis tools, wrote the paper.

Alfred E. Hartemink and Alex McBratney conceived and designed the experiments, wrote the paper.

Ho-Jun Jang performed the experiments, contributed reagents/materials/analysis tools.

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
