# Peer review of "Citations and the h index of soil researchers and journals in the Web of Science, Scopus, and Google Scholar"

_PeerJ, doi:10.7717/peerj.183_

## Round 0.1 · original submission · Minor Revisions

Thank you for a sound bibliometric study of the differences and relationships between h-indexes calculated by a variety of indexing services. There are some minor errors and issues with the use of English, which are covered comprehensively by the reviewers.

The only substantial change I would suggest is that you include a short discussion of the features of the soil science literature. The study is probably of interest to a broader audience and a consideration of the nature of this literature would go a long way to helping the reader judge its wider relevance. With this in mind, you might also consider replacing "pedobibliometric" with the more general "bibliometric" in the abstract and elsewhere.

If you have not already done so, please ensure that the data you used for the analysis are available, either as a supplementary file with PeerJ, or externally. If there is not a field-specific database of data available, there are a number of general purpose databases, including figshare.com and datadryad.com that you might use.

·

Basic reporting

The problem clearly stated and explained. The article flows nicely and the organization is easy to follow.

Experimental design

No comments

Validity of the findings

Check the paragraph beginning L239 "The Thompson Reuters Journal Citation Reports... " The first use of Soil Research as the journal name was February 2011. In 2010 the name was still Australian Journal of Soil Research, so I'm confused. This paragraph needs another look.

Additional comments

This paper is interesting to read with value to many researchers beyond soil scientists. Acknowledging  the differences between metrics from Web of Science, Google Scholar, and Scopus while demonstrating the relationship is interesting, and I am curious to see this applied in other disciplines. 

Add finesse to the conclusions by eliminating the bullet points. This section should flow like the rest of the paper. 

Catch the minor errors with a careful proofreading. Are the references in L220 and L255 for Table 6?

·

Basic reporting

There are some places where the English usage in the paper could be improved or where points being made need clarification. Detailed comments are:

Line 7 – The proper address protocol for the United States would be “…Madison, WI, 53706, USA.”

Line 14 – rewrite “…journal was being used…” to “…journal was used…”

Line 26 – rewrite “…seem to provided…” to “…seem to provide…”

Line 26 – rewrite “…higher number of…” to “…higher numbers of…”

Line 33 – “…mid-west…” should be “…Midwestern…”

Line 51 – Vine, 2006 is not in the references list.

Line 59 – rewrite “…citations is reliable…” to “…citations are reliable…”

Line 63 – suggest rewriting “…researchers with the following areas of interest in:…” to read “…researchers who listed their areas of interest as…”

Line 65 – rewrite “…Google Scholar’s author…” to “…Google Scholar author…”

Line 65 – rewrite “…author page.” to “…author pages.”

Line 66 – rewrite “…researcher was…” to “…researchers were…” (note both words are changed)

Line 69 – rewrite “…publication were inconsistent…” to “…publication record were inconsistent…”

Lines 76-77 – suggest rewriting “Table 1 shows the statistics of h index, number of publications, number of citations, and year of the first paper from 340 soil researchers in the three databases.” to read “Table 1 shows the h index, number of publications, number of citations, and year of the first paper statistics for 340 soil researchers in the three databases.

Lines 81-82 – Patel et al. is listed as 2013 in the manuscript, but as 2012 in the references. Which is correct?

Line 85 – rewrite “…of citation is…” to “…of citations is…”

Line 85 – rewrite “…Google scholar and the median is 866…” to “…Google Scholar, with a median of 866…” (note there are 5 changes in this line)

Line 90 – You state the periods are slightly different. I assume you mean the periods of time covered by each of the 3 databases? This should be clarified.

Line 96 – suggest rewriting “…1995 and revising the…” to “…1995 a revision was made to the…”

Line 96 – there should be a semicolon (;) after “authors”

Line 97 – rewrite “…20% extra..” to “…20% higher…”

Line 98 – rewrite “…finding by Falagas…” to “…findings of Falagas…”

Line 104 – rewrite “…relationship of number…” to ““…relationship between number…”

Line 114 – suggest rewriting “…citations will not affect much of the…” to “…citations do not have a large effect on the…”

Line 115 – rewrite “…citations the additional…” to “…citations additional…”

Line 115 – rewrite “… paper does not…” to “… paper do not…”

Line 118 – rewrite “…study from Franceschet…” to “…study by Franceschet…”

Line 121 – rewrite “…Scholar has five…” to “…Scholar had five…”

Line 122 – rewrite “…results of 340…” to “…results from 340…” Also, researchers is spelled wrong in this line.

Line 124 – rewrite “…for the h index Google…” to “…the h index from Google…”

Line 125 – rewrite “…is 1.1 higher…” to “…is 1.1 times higher…”

Line 127 – rewrite “…from UK…” to “…from the UK…”

Line 128 – Patel et al. is listed as 2013 in the manuscript, but as 2012 in the references. Which is correct?

Line 149 – rewrite “…publishing since 10 years…” to “…publishing for 10 years…”

Line 151 – suggest rewriting “…340 soil researchers (which…” to “…340 soil researchers in this study (which…”

Line 155-156 – There appears to be some misidentification of tables, and what the convey, in this section. In line 155 Table 4 is called for, but the information appears to come from Table 1. Then, in line 156, Table 1 is called for, but the information appears to come from Table 4. There should also probably be a call for Table 4 after the “0.7” in line 155.

Line 161 – rewrite “…ecology having the…” to “…ecology have the…”

Line 164 – rewrite “…consistent that soil…” to “…consistent in that soil…”

Line 165 – rewrite “…sub-discipline also varies in…” to “…sub-disciplines also vary in…” (note there are 2 changes)

Line 166 – delete the – after the :

Line 175 – rewrite “…low number of citations and the…” to “…low numbers of citations with the…”

Line 176 – rewrite “…decreases with…” to “…decreasing with…”

Lines 182-184 – There is a discussion here of how many citations the average soil researcher receives from various types of publications. Is this the average number of citations per year? The time interval needs to be clarified within this sentence.

Line 183 – rewrite “…journal articles, additional…” to “…journal articles, an additional…”

Line 199 – “…references quoted…” Wouldn’t it be more accurate to say “…references cited…”?

Line 204 – rewrite “…percentage self-citation…” as “…percentage of self-citation…”

Lines 204-205 – It is stated here that established authors cite less of their own work, and that is based on the fact that a weak relationship was found between % of self-citation and scientific age. But the analysis was based on % self-citation. If a young author has 400 citations, and 40 of them are self-citations, they have a self-citation rate of 10%. However, if an established author has 8000 citations, and 400 of them are self-citations, that established author has cited themselves as many times as the young author has total citations, but has a lower self-citation % (5%, or half as much on a % basis). Therefore, I don’t think it is accurate to stay in your paper “…established authors cite less of their own work.” In the example above, the established author has cited more of their own work than the young author. What is true is that a lower % of the citations received by established authors is due to self-citation, and that statement would work in your manuscript.

Line 207 – rewrite “…researcher matures the papers…” to “…researcher matures their papers…”

Line 208 – suggest changing “…and more citations…” to “…and more external citations…”

Line 214 – rewrite “…of journal’s metric, h5-index…” to “…of a journal’s metrics, the h5-index…” (note 3 corrections in this line)

Line 215 – rewrite “…journals for soil…” to “…journals for the soil…”

Line 219 – rewrite “…Scholar metric…” to “…Scholar metrics…” (metrics is plural on the Google Scholar page)

Line 220 – The manuscript refers to Table 5, but the material referenced appears to be in Table 6.

Line 221 – rewrite “…IF, whereas Figure…” to “…IF, and Figure…”

Line 222 – rewrite “…year IF has a…” to “…year IF have a…”

Line 225 – rewrite “…different in the ranking…” to “…different in their ranking…”

Line 226 – rewrite “The 2 journals are…” to “The top two journals are…”

Line 226 – rewrite “…maintain high number…” to “…maintain a high number…”

Line 227 – rewrite “…papers they published.” to “…papers they publish.”

Line 233 – rewrite “All these are journals…” to “All these journals…”

Line 235 – rewrite “…GS and ranked 25 in…” to “…GS and 25 in…”

Line 248 – rewrite “…increase much rapidly with…” to “…increase rapidly with…”

Line 249 – rewrite “…1000 citation). For the other 7 journals…” to “…1000 citations). For the seven journals…”

Line 250 – rewrite “…is half (0.6 IF increase…” to “…is half as much (0.6 IF increase…”

Line 251 – rewrite “…for journals to publish…” to “…for journals that publish…”

Line 251 – rewrite “Meanwhile h5-index…” to “Meanwhile the h5-index…”

Line 252 – rewrite “…by no. citations…” to “…by no. of citations…”

Line 253 – rewrite “…from WoS, h5-index…” to “…from WoS, the h5-index…”

Line 255 – Table 5 is referenced but the material seems to be in Table 6.

Line 255 – rewrite “…that GS h5-index…” to “…that the GS h5-index…”

Line 258 – rewrite “Citation from a highly…” to “A citation from a highly…”

Line 260 – rewrite “…self-citation as compared to impact…” to “…self-citation than the impact…”

Lines 267-268 – “…and is shows that only 1-9% (median 5%) of the total papers that contributed to h5-index.” This is an incomplete thought. What about the 1-9% of the total papers that contributed to the h5-index?

Line 268 – rewrite “And we also…” to “We also…”

Line 268 – rewrite “…that h5-index keeps…” to “…that the h5-index keeps…”

Line 270 – rewrite “…use of GS h index…” to “…use of the GS h index…”

Line 272 – rewrite “…performance when compared to impact…” to “…performance than the impact…”

Line 275 – Should Falagas and Alexiou, 2008 be Falagas et al., 2008? There is a Falagas et al. in the references list, but no Falagas and Alexiou.

Line 277 – rewrite “…artificially increases the…” to “…artificially increase the…”

Line 288 – rewrite “…of citation and…” to “…of citations and…”

Line 290 – rewrite “…relationships of the h index between these…” to “…relationships between the h indices in these…”

Line 294 – rewrite “…affects for assessing scientific…” to “…affects the assessment of scientific…”

Line 299 – rewrite “…Scholar include other…” to “…Scholar includes other…”

Line 301 – rewrite “…anyone interested…” to “…anyone interested in…”

Line 302 – rewrite “…aware whether these…” to “…aware of whether these…”

Line 304 – rewrite “…relating h index…” to “…relating the h index…”

Lines 316-318 – Should this be Falagas et al., 2008 or Falagas and Alexiou, 2008?

Lines 351-354 – Should this be Patel et al. 2012 or 2013?

Line 357 – Vine, 2006 should be entered here.

Table 1 – The right side of my Table 1 has been cut off.

Experimental design

The experimental design and statistical analyses are solid.

Validity of the findings

The findings are supported by the statistical analysis.

Additional comments

This is an interesting study that sheds light on how we rate, or evaluate, the prestige of scientific work done by individual researchers and the journals that publish that work in the soil science discipline. Such studies are important to fully understand the reporting of scientific findings and motivations behind the choices scientists make in selecting publication outlets.

Reviewer 3 ·

Basic reporting

I think that it is a useful bibliometric study, which would be useful for all the specialists in scientific planning, and especially for soil scientists. The only addition I would suggest is a short discussion of the peculiarity of publication in soil science (generally it is a longer period between data collection and publication, relatively small community etc.). I think that a couple of sentences in the introductory part would be great.

Experimental design

No comments.

Validity of the findings

No comments.

---

## Round 0.2 · accepted · Accept

Thank you for your comprehensive treatment of the reviewers' and my comments. The final manuscript should be relevant to a broader audience than was originally the case.